# Predicting firm creation in rural Texas: A multi-model machine learning approach to a complex policy problem

Mark C. Hand[1,2]*, Vivek Shastry[2], Varun Rai[2]

1 University of Texas at Arlington, Arlington, Texas, United States of America, 2 LBJ School of Public Affairs, University of Texas at Austin, Austin, Texas, United States of America

☯ These authors contributed equally to this work.

* mark.c.hand@gmail.com

## Abstract

Rural and urban America have becoming increasingly divided, both politically and economically. Entrepreneurship can help rural communities catch back up by jumpstarting economic growth, creating jobs, and building resilience to economic shocks. However, less is known about firm creation in rural areas compared to urban areas. To that end, in this paper we ask: What factors predict firm creation in rural America? Our analysis, based on a comparative framework involving multiple machine learning modeling techniques, helps addresses three gaps in academic literature on rural firm creation. First, entrepreneurship research stretches across disciplines, often using econometric methods to identify the effect of a specific variable, rather than comparing the predictive importance of multiple variables. Second, research on firm creation centers on high-tech, urban firms. Third, modern machine learning techniques have not yet been applied in an integrated way to address rural entrepreneurship, a complex economic and policy problem that defies simple, monocausal claims. In this paper, we apply four machine learning methods (subset selection, lasso, random forest, and extreme gradient boosting) to a novel dataset to examine what social and economic factors are predictive of firm growth in rural Texas counties from 2008–2018. Our results suggest that some factors commonly discussed as promoting entrepreneurship (e.g., access to broadband and patents) may not be as predictive as socioeconomic ones (age distribution, ethnic diversity, social capital, and immigration). We also find that the strength of specific industries (oil, wind, healthcare, and elder/childcare) predicts firm growth, as does the number of local banks. Most factors predictive of firm growth in rural counties are distinct from those in urban counties, supporting the argument that rural entrepreneurship is a distinct phenomenon worthy of distinct focus. More broadly, this multi-model approach can offer initial, focusing guidance to policymakers seeking to address similarly complex policy problems.

**Data Availability Statement:** All relevant data are within the paper and its Supporting information files.

**Funding:** This study was funded by a grant to VR by the IC2 Institute at the University of Texas at

Austin (https://ic2.utexas.edu/). The funders had no role in study design, data collection and analysis, decision to publish, or preparation of the manuscript.

**Competing interests:** The authors have declared that no competing interests exist.

## Introduction

In the United States, the years following the Great Recession of the late 2000s is a tale of two recoveries. While urban areas saw slow, steady recovery, in the run-up to the 2016 election, some rural states had yet to return to their pre-Recession levels employment [1]. Closing that gap means building the economies of rural areas; this, in turn, depends on higher rates of entrepreneurship and self-employment, which can help communities jumpstart and sustain economic growth, create jobs, and mitigate trade shocks more effectively than attracting large firms [2]. For rural areas in particular, promoting entrepreneurship may be one of few effective economic development strategies [3]. In these areas as elsewhere, spurring entrepreneurial activity can launch a larger feedback loop, reinforcing the regional conditions that make for further firm creation feasible [4].

But what factors are predictive of firm creation in rural America? Despite the importance of this question to policymakers, we know little about firm creation in rural areas compared to urban areas. According to one recent report, "More research is needed on what types of entrepreneurs are effective in promoting growth and on what types of infrastructure and technical-assistance support (including business planning and other forms of education) are most effective in helping develop and grow local businesses" [2]. This is particularly true in the U.S., given that most of the work on entrepreneurial rate sis based in Europe or the developing world. One recent review found that rural entrepreneurship is "an essentially European concern" [5]. This is only just beginning to shift, with U.S. scholars calling for greater focus on more "ordinary" startup firms [6] and others putting forward typologies of state-level development interventions [7].

As interest in rural entrepreneurship has grown (See Fig 1), scholars of rural economic development have relied on existing entrepreneurship literature for clues about what factors might predict firm creation in rural areas specifically. In Table 1, we gather the most well-cited scholarship on entrepreneurial rates, collected from a range of disciplines and theoretical perspectives, and divided into natural, market, cultural, demographic, and regulatory determinants of firm creation.

Drawing from the broad theoretical categories in Tables 1 and 2 outlines the results of classic and recent empirical work on firm creation, which have identified the following variables to be associated with firm creation in urban areas: market characteristics (the availability of financial capital, labor and suppliers; the absence of an oil and gas industry), demographic characteristics (ethnic diversity, population density, age distribution, immigration); regulatory policies (democracy, control of corruption, labor market freedom, low interest rates, property rights, R&D expenditures and incentives programs [negatively correlated]), cultural characteristics (risk-taking, gender-egalitarianism, performance reward, and the social status accorded to risk-taking entrepreneurs), and even the amount of recent bridge construction [18]. Except for one study, much of the research till date has not placed adequate emphasis on entrepreneurship in rural counties.

Based on a review of the analytical approaches and variables used in the related literature as presented in Tables 1 and 2, we see three important gaps in this existing literature, each of which we address in this paper. First, the list of factors that contribute to firm creation is expansive but theoretically disconnected, coming from multiple and often contradictory epistemological perspectives. Researchers with multiple lenses on entrepreneurship have yet to pull together a theoretically-coherent picture of the socio-economic factors that lead to firm growth [5, 19]. This theoretical complexity, according to one recent review, is reflected in the existence of at least six distinct, disconnected theoretical approaches to rural entrepreneurship

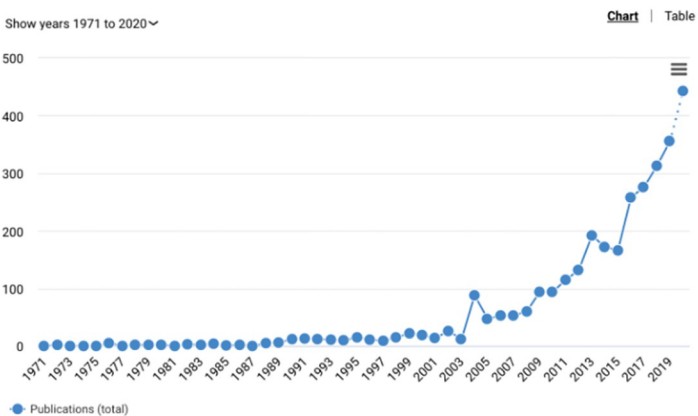

**Fig 1. Scholarly papers including the phrase *rural entrepreneurship*.** Source: Dimensions.ai, accessed 18th November 2020.

[20]. That theoretical disconnection has hampered progress in understanding the causes and effects of rural entrepreneurship.

The second gap in existing research on determinants of entrepreneurial activity is that existing work on rates of entrepreneurship mostly emerged from the study of entrepreneurship in urban areas. Much of what we know so far about entrepreneurship is focused on high-growth, big-city entrepreneurs to be relevant in rural areas, in which entrepreneurship and innovation

**Table 1. Previously theorized determinants of firm creation by category.**

| Natural | Market | Cultural | Demographic | Regulatory | Source |
|---|---|---|---|---|---|
| Natural cost advantages | Agglomeration of customers, suppliers, labor, and Technology spillovers | Entrepreneurial culture | Demographics | - | [8] |
| - | - | Social norms-culture, Cognitive dimensions, Belief systems | - | Political structure, Procedures-regulations, Contracts, Property rights | [9] |
| - | Market Conditions, Creation and Diffusion of Knowledge, Access to Finance | Entrepreneurial Culture | Entrepreneurial Capabilities | Regulatory Framework | [10] |
| - | Technological change, Per capita income, Savings, Demand, Unemployment rates, Wage rates, Failure rates | Protestant ethic, Existing innovative firms | - | Political change, Tax rates, Low interest rates | [11] |
| - | Density of established organizations, Technology transfer to new companies, Venture capital (VC) availability | Gender-equity, value, and reward norms; status endorsement | Quality of STEM education | National investment in R&D, Governmental support and policies for entrepreneurship | [12] |
| - | Technology Absorption, Globalization, Competition, Financing | Cultural Support | Human Capital | - | [13] |
| - | Financing for Entrepreneurs, Commercial and Professional Infrastructure, Internal Market Dynamics, Internal Market Openness | Cultural and Social Norms | Entrepreneurial Education and Training | Government Support, Policies & Programs, Taxes & Bureaucracy, R&D, Physical & Services Infrastructure | [14] |
| Sensitivity to weather | Size of firms, growth orientation, subsidy dependence, agglomerative effects, transportation & information costs, cognitive proximity to new technologies | Strength of weak ties | Labor productivity | - | [15] |
| - | Self-employment earnings | Culture favoring entrepreneurship | Population density, ethnic diversity | State policy toward labor market freedom | [16] |
| - | Market Conditions, Access to Finance, Creation and Diffusion of Knowledge, | Entrepreneurial Culture | Entrepreneurial Capabilities | Regulatory Framework | [17] |

**Table 2. Variables associated with firm creation, across contexts.**

| Regional Determinants | Context | Source |
|---|---|---|
| Changes in technology, Protestant Ethic[1], interest rates, prior rates of entrepreneurship, risk-taking propensity, business failure rates, economic growth, immigration, age distribution of population | US firm creation at the national level, 1900s | [11] |
| Financial entrepreneurial capital (inverse U-shaped relationship) | US states | [23] |
| Abundant workers and presence of many small suppliers; to a lesser extent level of local customers and suppliers | US manufacturing startups across cities and industries | [8] |
| Ethnic diversity of the population, population density, state-level labor market freedom policy | Self-employment in US counties | [16] |
| Growth in nearby MSAs (-) | Self-employment in rural US counties | [24] |
| Incentives (-) and inter-sectoral job mobility | Post-recession start-up rates in US counties | [25] |
| Oil and gas sector expansion (-) | Self-employment in US counties | [26] |
| Norms of gender-egalitarianism, value and reward of performance, and endorsement of status privileges | Company founding across countries | [12] |
| Access to stock markets and the financial system, hiring and firing rules and controls, control of corruption, democracy, government size and capability, property rights, the presence of role models | Multiple | [9] |
| Innovation and new technologies, peer effects, the sociocultural environment, R&D transfers, and the availability of government subsidies | Entrepreneurial founding across countries, 2014 | [27] |

[1]We consider this variable to be culturally biased and irrelevant for this analysis. Its insignificance as a determinant of firm creation is further evident from our results.

Note: A minus sign (-) indicates a negative correlation

are focused on agriculture, tourism, decentralized service provision, and manufacturing [21], and where entrepreneurial ventures may take different, perhaps more slow-growth, forms. Just as rural counties in the United States experienced a significant divergence in economic indicators from urban counties in the wake of the Great Recession [22], so might there might be material differences in the ways that rural and urban entrepreneurs respond to changes in their environment.

The third gap in existing research is methodological. As in other complex socioeconomic systems, the effects of all the variables identified in Table 2 on firm creation are difficult to isolate because they are by their nature collinear and mutually reinforcing. That complexity outstrips the ability of standard econometric tools to help us understand such a system beyond basic correlations. When the number of potential variables becomes large, or when their interactions are important but unknowable, predictive methods can fill the gap—not by establishing causal connections between one independent and dependent variable, but by surfacing the variables that merit the greatest research focus [28].

Like many disciplines, policy research has depended greatly on causal inference through econometric methods. As Kleinberg et al. argue, this makes sense, given that policymaking often requires consideration of a counterfactual; what will happen with and without a particular policy? [28] Not all policy questions are structured like this, however, and especially not at the beginning of the agenda-setting process. During the Covid-19 crisis, for example, the first task rural policymakers faced was what outcomes they should prioritize [29]. Having established their priorities, policymakers need to consider multiple dimensions of policy responses that have complex interactions and tradeoffs [30]. When the question at hand is "among the hundreds of possible policy responses, which should we consider further?" then the task is to

establish predictive accuracy first, then attempt to identify the relationships among the predictors [31].

The use of machine learning methods has greatly expanded in public policy scholarship over the last decade. In criminology, the first policy discipline in which machine learning began to be incorporated in 2013 to forecast reoffending [32], researchers have used it to predict local crime rates [33], partner violence [34], firearm violence and mass shootings [35], sexual recidivism [36], and land use policies [37]. Even in 2013, the outlines of the dangers of its application—to reinforce discriminatory institutions or provide new tools to authoritarian governments, e.g.—were clear, having been previewed in other disciplines [38, 39]. Since then, the use of machine learning in public policy has ramified into other policy domains, including poverty-reduction, workforce development and health inspections [for a more complete list, see [28, 40].

Entrepreneurship and innovation scholars are working even more assertively in this predictive direction, with recent reviews focused on the potential applications of "big data" and artificial intelligence to entrepreneurship research [41, 42]. Gimenez-Nadal et al. [27] apply bootstrap and forward selection techniques to the Global Entrepreneurship Monitor data for the year 2014, for example, finding that "innovation and new technologies, peer effects, the sociocultural environment, entrepreneurial education at University, R&D transfers, and the availability of government subsidies are among the most important predictors of entrepreneurial behavior." [43] use Italian firm data to predict how innovative firms will be, and then correlate that measure with subsequent firm survival. [44] uses stacked generalization and boosted trees to predict the size of venture financing rounds. [45] use multiple machine algorithms to predict venture success, and [46] to suggest successful revenue models.

This paper addresses each of those gaps in prior research, with three research objectives. First, we aim to provide an answer to the question of what factors predict firm creation in rural America. Second, we aim to demonstrate how those factors differ from urban America. Third, we aim to demonstrate how policy scholars and the policymakers they advise can deploy computational methods to complex policy problems without monocausal explanations.

To accomplish this, we pull together variables identified by multiple disparate strands of rural entrepreneurship research into machine learning models to identify what factors predict the of firm growth in 254 Texas counties across ten years. We also identify how the factors that predict rural entrepreneurship differ from the factors that predict urban entrepreneurship, both in prior research and in our own models, as we expected.

In the next section of this paper, we describe what data we collected, and how. We then describe each of the four machine learning models we deployed, why we chose them, and how we combine the results of those models into a single output that policymakers can make sense of. We then share the results of each individual model of rural firm creation, the combined results, and a comparison of those results to our urban firm creation models.

In the discussion, we interpret those results, including their implications for rural entrepreneurship scholars and policymakers alike. We then consider how policy scholars and practitioners might apply this approach to other complex policy issues.

## Data

In the analyses below, we examine firm creation in all 254 counties in Texas across ten years, from 2008–2018, a period in which 1,255,963 new firms were created in the state. According to the U.S. Office of Management and Budget, 157 of those counties are rural, and the rest are urban or micropolitan. Together, these counties account for 8.4% of all U.S. counties and 8.7% of the total U.S. population, providing variation across a wide dimensional space but allowing

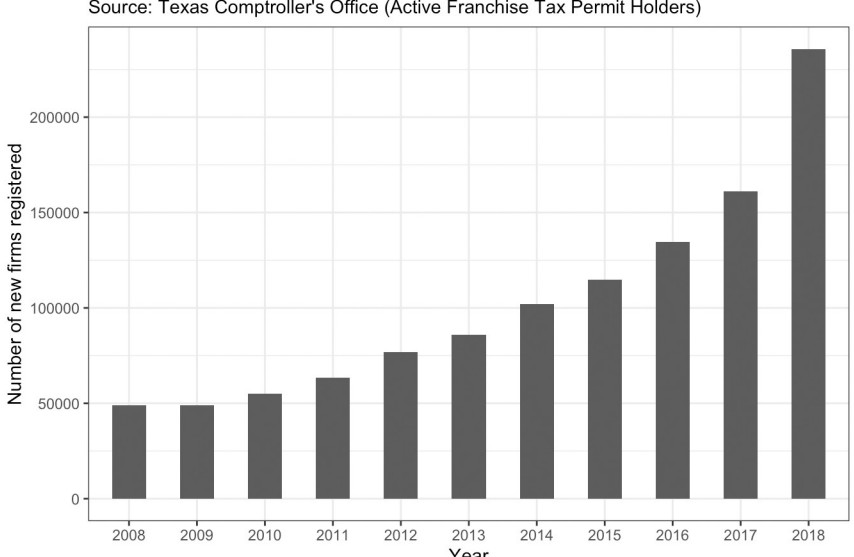

Source: Texas Comptroller's Office (Active Franchise Tax Permit Holders)

**Fig 2. New firms registered in rural Texas from 2008–2018.**

us to control for state policy regime. Texas also has a unique publicly available dataset on firm creation: Texas Comptroller of Public Accounts' Active Franchise Tax Permit Holders data. Within this data, which stretches from 1843 to the present, each business in Texas is assigned a *Responsibility Beginning Date* from which we build our dependent variable of county-year firm creation, our dependent variable. Fig 2 shows the number of firms created in rural Texas from 2008–2018, our period of study; see Appendix A in S1 Appendix for descriptive statistics of all variables. This unique data frame limits our current study to Texas counties, and the availability of our independent variables listed below limit our study to 2008–2018. We return in the discussion to the question of generalizability from this data, in the context of the decision-making process of public policymakers.

All data used in this analysis are available in a reproducible RStudio Project uploaded to protocols.io, available at https://www.protocols.io/file-manager/6AE6BC0F7C8C11EDA3 DF0A58A9FEAC02.

Our list of independent variables is built from a range of data sources, including the US Census's County Business Patterns, the US Geographical Survey wind turbine database, the University of California Berkeley Lab's Tracking the Sun database, and the Texas Railroad Commission by way of Texas2036, an Austin-based think tank. In line with a report prepared for the Appalachian Regional Commission [47], we construct multiple variables from the US Census's American Community Survey. We use lending data from the Community Reinvestment Act, bank availability data from the Federal Deposit Insurance Corporation, patent data from the US Patent Office, broadband access data from the National Telecommunications and Information Administration, and several variables from the Robert Wood Johnson Foundation's County Health Rankings. Many of these variables are unavailable before 2008, so we restrict our time series to 11 years from 2008–2018. In Table 3 we list those variables and their sources.

As predictor variables, we use the datasets above to build measures that account for just over 40 percent of the variables used by previous researchers as listed in Table 2, either directly or by close proxy. The remaining variables are not available at a county-level resolution or are

**Table 3. Description of variables and sources.**

| Variable Type | Variable ID | Description | Source | Timeframe |
|---|---|---|---|---|
| Dependent Variable | New Firms | Number of new firms established per county per year | Texas Comptroller's Active Franchise Tax Permit Holders | 2008–2018 |
| Demographic Variables | Population | Total population | American Community Survey | 2008–2018 |
| | Population density | Population per land area of the county | American Community Survey | 2008–2018 |
| | Pop. aged 25–44 (%) | Percent of population aged 25–44 years (%) | American Community Survey | 2008–2018 |
| | Pop. aged 65 and older (%) | Percent of population aged 65 and above (%) | American Community Survey | 2008–2018 |
| | Pop. born in state (%) | Percent of population who were born in the state of residence (%) | American Community Survey | 2008–2018 |
| | Pop. college education (%) | Percent of population age 25+ with bachelor's degree or higher (%) | American Community Survey | 2008–2018 |
| | Ethnic diversity | Ethnic diversity of minorities (except white) | American Community Survey | 2008–2018 |
| | Immigration (%) | Percent of population who are in-migrants (%) | American Community Survey | 2009–2015 |
| | Outmigration (%) | Percent of population who are out-migrants (%) | American Community Survey | 2009–2015 |
| | Protestant Ethic | Ratio of Protestant to Catholic residents | The Association of Religion Data Archives | 2010 |
| Economic Variables | Unemployment rate | Unemployment Rate | Bureau of Labor Statistics | 2008–2018 |
| | Self-employment | Ration of (nonfarm) proprietors to total employed | Bureau of Economic Analysis | 2008–2018 |
| | Total Employment | Total employment in all occupation categories | Bureau of Economic Analysis | 2008–2018 |
| | Banks | Number of banks per county | Federal Deposit Insurance Corporation | 2008–2018 |
| | Bank deposits | Total deposits held in local banks, $1000s | Federal Deposit Insurance Corporation | 2008–2018 |
| | Federal Funds Rate | Effective Federal Funds Rate | Federal Reserve Economic Data | 2008–2018 |
| | Income inequality | Gini index of income inequality | American Community Survey | 2008–2018 |
| | Pop. in poverty (%) | Percent of population in poverty (%) | American Community Survey | 2008–2018 |
| | County GDP | Current-dollar GDP (thousands of current dollars) | Bureau of Economic Analysis | 2008–2018 |
| | Failed firms (prior year) | The number of firms that closed in the prior year. | Texas Comptroller's Office | 2008–2018 |
| | New firms (prior year) | Number of new firms established in the previous year (lagged DV) | Texas Comptroller's Office (Active Franchise Tax Permit Holders) | 2009–2018 |
| | business_density | Number of establishments per 1,000 age 20–64 population | County Business Patterns | 2008–2018 |
| | Business density | Number of patents issued in each county | U.S. Patent Office | 2008–2018 |
| | Percent insured | Percent of population with health insurance | American Community Survey | 2008–2018 |
| | Broadband access | Percent of population with access to broadband internet | Federal Communications Commission | 2016–2017 |
| Energy Industry Variables | Oil production | Total gas production (barrels) | Railroad Commission of Texas (curated by Texas 2036) | 2008–2018 |
| | Gas production | Total gas production (barrel of oil equivalents), including casinghead gas production | Railroad Commission of Texas (curated by Texas 2036) | 2008–2018 |
| | Solar installations | Total solar power installed Capacity (in MW) | University of California Berkeley Lab (Tracking The Sun database) | 2008–2018 |
| | Wind capacity | Total number of wind turbine installations | U.S. Geological Survey (Wind Turbine Database) | 2008–2018 |
| | Oil play | Which of the five major oil and gas plays a county is part of | Texas Railroad Commission | 2020 |
| Other Industry Variables | Industry size: farming | Percent of employment in farming—NAICS 11 (%) | County Business Patterns | 2008–2018 |
| | Industry size: extraction | Percent of employment in extraction—NAICS 21 (%) | County Business Patterns | 2008–2018 |
| | Industry size: recreational | Percent of employment in arts, entertainment, and recreation—NAICS 71 (%) | County Business Patterns | 2008–2018 |
| | Industry size: oil and gas | Percent of employment in oil and gas extraction (%) | County Business Patterns | 2008–2018 |
| | Industry size: K-12 education | Percent of employment in Elementary and Secondary Schools—NAICS 6111 (%) | County Business Patterns | 2008–2018 |

*(Continued)*

**Table 3.** (Continued)

| Variable Type | Variable ID | Description | Source | Timeframe |
|---|---|---|---|---|
| | Industry size: manufacturing | Percent of employment in manufacturing—NAICS 31 (%) | County Business Patterns | 2008–2018 |
| | Industry size: community college | Percent of employment in Community college—NAICS 611210 (%) | County Business Patterns | 2008–2018 |
| | Industry size: healthcare | Percent of employment in offices of physicians, dentists, and other health practitioners—NAICS 6211–3 (%) | County Business Patterns | 2008–2018 |
| | Industry size: coal | Percent of employment in coal mining—NAICS 2121 (%) | County Business Patterns | 2008–2018 |
| | Industry size: elder and childcare | Percent of employment in child day and elderly care services—NAICS 62441, 61412 (%) | County Business Patterns | 2008–2018 |
| County-Level Variables | Land area | Land area in square miles | ARC 2018 (Cited from U.S. Census Bureau) | 2000 |
| | Distance from metro area | Distance (miles) to a county with more than 250,000 population | ARC 2018 (Author's calculation) | 2005 |
| | Natural Amenities score | Natural amenities | ARC 2018 (Cited from U.S. Dept. of Ag) | 2005 |
| | 2020 presidential vote differential (%) | Difference between two major party presidential candidates | MIT Election Lab | 2020 |
| | Social capital | Social Capital Index | Northeast Regional Center for Rural Dev. | 2014 |

constructed from proprietary datasets, which we avoid to improve transparency and reproducibility of our analysis. We then add several variables not included in previous models, including wind capacity and solar installations, two other growing sources of energy production and employment in rural Texas. As additional economic indicators we include unemployment rate, income inequality, and industry diversity. In addition to migration variables, we create and add a variable for ethnic diversity. We also include a measure of counties' ability to regain job growth after the Great Recession (called resilience by Boettner et al.), along with a number of variables for specific industries associated with resilience [47].

Most of the variables used in the analysis below are available at the county-year level for each of those years. Where the data for a certain variable are not available for every year, but multiple values were available over the period (for example the percentage of population who were in- or out-migrants), we impute values so as not to lose these observations during analysis. For those variables where only one value is available (for example social capital index), we attribute that value as a stable characteristic of the county. Lastly, to be able to compare the magnitude of coefficients across variables and the predictive accuracy across models, we standardize all variables.

## Methods

In modeling studies results may be sensitive to the data and the modeling technique selected. In econometrics this issue is partly addressed by conducting sensitivity analyses by modifying parameter specifications while typically keeping the core algorithm unchanged. While the strength of machine learning methods is in being able to implement non-parametric non-linear algorithms to achieve higher predictive accuracies, there is often less visibility into the inner workings of the algorithms. Therefore, it is possible that the results are sensitive to the specified machine learning algorithm. This issue is widely recognized, and in many fields, it is now standard practice to implement a number of different models on the same dataset to compare their predictive accuracies [48–51]. Random forest models can, for example, be used to predict rare events such as the onset of civil wars with much higher accuracy than classical

logistical regressions models [52]. This is just one example, of course; in other comparisons linear or logistics models might perform better than more flexible algorithms. *Ex ante*, it is impossible to know what sort of model performs best on a set of data.

The complex algorithms that machine learning models employ in pursuit of greater predictive accuracy often trade-off interpretability. When the modeling objective is to optimize the algorithms to achieve the highest predictive accuracy, models that achieve the lowest prediction errors can be selected, regardless of their interpretive value. However, while investigating complex social science problems such as we do in this paper, interpretability is as important as predictive accuracy. While predictive accuracies give an understanding of the relative performance of different models, different models offer different types of interpretive capabilities, and there is no straightforward way to compare the interpretation of variables from different models. This is important, because two or more models may perform almost equally well, but because of their use of different algorithms, they may display completely different relationships between the dependent and independent variables [31]. In any case, multiple methods might actually be necessary to investigate complex causal relationships [53]. We therefore argue in favor of and propose a framework for comparing the importance of variables across multiple machine learning models. This allows for an exploration of variables that appear consistently across different models, and those that are sensitive to model-specific algorithms.

## Model selection

Linear regression is the simplest and most popular modeling technique that finds the line of best fit by minimizing the sum of squared residual errors. The strength of linear regression lies in the easy interpretability of resulting coefficients. Among the many widely known limitations of linear regressions, an important one is its focus on minimizing in-sample error, which can lead to problems of "over-fitting" and poor performance while predicting out-of-sample data. For improving the predictive capability of a model, it is important to reduce the out-of-sample error, even at the expense of trading off in-sample error. Machine learning techniques offer an empirical way to manage this trade-off and improve out-of-sample predictive performance [54]. Specifically, in this paper we use 10-fold cross validation technique to obtain the out-of-sample prediction errors (also called Test Mean Standard Error or Test MSE) for comparing the performance of different models.

In this paper, we use linear regression, linear regression with forward and backward subset selection, lasso regression, random forest, and XG Boost models to investigate the socio-economic determinants of new firm starts in Texas counties. We have selected these techniques for their ability to model a continuous response variable, as well as their ability to provide interpretable outputs, albeit in different forms. Other models such as K-nearest neighbor (KNN) regression and Principal Component Regression can also be applied but offer less interpretable results. A number of textbooks such as [55] provide a detailed introduction to a range of machine learning techniques that may be applied over the interpretability vs. complexity tradeoff spectrum.

Forward and backward subset selection models work by iteratively adding (or removing) variables from a series of models, one-at-a-time, at each step considering which variable offers the greatest additional improvement (or reduction) in model fit. From this series of models, at each step the one with the least Bayes Information Criterion (BIC) statistic is selected as the best performing model, thus yielding a "greedy" (i.e., suboptimal) but computationally efficient approach. Lasso regression belongs to a class of "shrinkage" or "regularization" models that apply a tunable penalty term while minimizing the residual sum of squares, resulting in a sparse model wherein less important coefficients could be set to zero. Random forest is a class

of tree-based models that aggregates repeated, decorrelated decision trees build from random subsets of predictors at each step. Finally, extreme gradient boosting, or XG Boost, is a technique that uses the processes of gradient boosting and regularization to improve the prediction rate by adjusting the weights and penalties for errors made by previous weaker models.

The random forest, lasso regression, and XG boost models were tuned to optimize the hyperparameters to minimize the out-of-sample errors, as is standard in the literature [56–58]. The random forest model was tuned to select the optimal mtry value (where mtry is the number of variables randomly sampled at each split). In the lasso regression model, cross validation was used to select the lambda value which minimizes the mean cross-validated error. In the XG boost model, and in line with prior research, a grid search with 5-fold cross validation was performed to tune five hyperparameters—the maximum depth of a tree (max_depth), subsample ratio of the training instance (subsample), subsample ratio of columns when constructing each tree (colsample_bytree), lambda, and alpha values [59–62].

In terms of outputs, linear regressions provide the magnitude and direction of the coefficients associated with each predictor, along with p-values indicating their level of significance. Subset selection and lasso regression models result in a selected subset of predictors along with their magnitude and direction of impact. Random forest models generate variable importance plots, which is a summary of the extent to which each predictor contributes to the reduction in mean squared error and gini index, which are both indicators of their relative level of importance.

We chose these techniques for their ability to model a continuous response variable, as well as their ability to provide interpretable outputs, albeit in different forms. Other models such as K-nearest neighbor (KNN) regression might be used but offer less interpretable results. Many other techniques such as Principal Component Regression, Support Vector machines, Linear and Quadratic Discriminant Analyses and other tree-based techniques can be applied based on the nature of the research questions and the underlying data. Several textbooks such as (James et al., 2013) provide a detailed introduction to these machine learning techniques.

## A multi-model interpretation framework

Each machine learning method provides a different type of output to assist in interpreting the importance of predictors. The strength and relative importance of predictors might vary across different methods, and models may themselves have relatively different predictive capabilities. To assess the overall contribution of each predictor from across all the models considered, we develop a novel combined interpretation framework that allows for systematic aggregation of outputs from multiple models. The rationale for this combined interpretation framework is to achieve three objectives: (a) prioritize variables with higher magnitude of association within each model, (b) prioritize variables that appear as significant across multiple models, and (c) prioritize variables from models that have relatively better predictive performance.

The combined interpretation framework consists of three steps (Fig 3). First, we take the list of variables from each model output and rank them according to their importance. The ranking scheme depends on the nature of output from each model. In the random forest model, we rank predictors according to their contribution to reducing the percent mean-

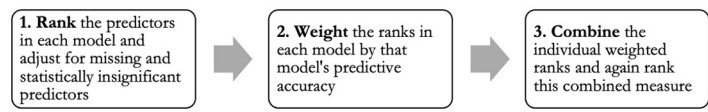

**Fig 3. A three-step framework for interpretating of results of multiple ML models.**

squared error. In all other models, we rank the predictors according to the magnitude of their (standardized) coefficients. This ensures that the predictors that are more important in each model get higher ranks. Further, the predictors missing in each model and predictors that are not statistically significant in the linear regression model are penalized by assigning their rank as one greater than the total variables in the largest model. This ensures that the predictors that are not significant in each model get lower ranks.

Second, once the rank for each predictor is assigned for all models, these ranks are weighted by the respective model's predictive accuracy. In our case for example, we weight the ranks from each model by that model's 10-fold cross validation error. This ensures that the predictors in better performing models get higher weightage.

Finally, we combine the weighted ranks from all models. To do this, we take the average rank of each predictor across all models. Then, we rank those averages. This ensures that the predictors appearing across multiple models get higher ranks. This is the final measure of each predictor's overall importance in our models, as described below.

## Results

The abbreviated output of the models is shown in Table 4, and the cross-validated mean squared errors (CV Error) of each of our five models are shown in Table 5 (see Appendix B in S1 Appendices for a list of variable appearances across all models; and Appendix C in S1 Appendices for hyperparameter tuning plots). The linear model performed best, followed in order by the XG Boost, random forest, subset selections and lasso regression models. In many circumstances, the process of interpreting machine learning outputs might stop here: Having discovered the best model(s), we would simply interpret the results of the best-performing model. However, there is no guarantee *ex-ante* that this model always produces the least prediction errors. The tuning parameters in each model and the cross-validation errors, among other steps, are all a function of the underlying data and stochastic processes. With a different randomization routine or with a different dataset, it is entirely possible that a different model performs better [63], especially when two or more models perform closely overall such as the linear regression and random forest models in this paper. Therefore, we implement the three-step framework described above to obtain a combined single rank of importance of all variables across all models. The resulting table is shown in Table 6.

### County characteristics and demographics

Prior research has suggested that high rates of firm creation are associated with a number of demographic and cultural characteristics of a local population, including its age distribution [11], education levels [14, 64], ethnic diversity [16], culture [12], social capital [65], population growth [66], rates of immigration and age of immigrants [67], population density [16], and (oddly) the ratio of Protestant to Catholic religious adherents [11]. In the consolidated results in Table 6, some of these stand out, including age distribution, ethnic diversity, social capital, and migration, in descending order of predictive variable importance.

One other factor merits discussion here: the number of new firms started in the previous year. There are at least two ways to interpret that the number of firms started in the prior year is predictive of number of firms started in this year. First, we might understand it structurally: Those factors that are associated with higher entrepreneurial rates carry over from year to year. Second, we might understand it culturally, as many of the authors in Table 1 theorize: Once an entrepreneurial culture is established in a place, it becomes more culturally appropriate to take entrepreneurial risk [27].

**Table 4. Comparing features across models.**

| Variable[2] | RANDOM FOREST | | Multilinear regression[1] | | Forward subset | | Backward subset | | Lasso | | XG boost |
|---|---|---|---|---|---|---|---|---|---|---|---|
| | % inc MSE | Inc node purity | Coef. | +/- | Coef. | +/- | Coef. | +/- | Coef. | +/- | Gain |
| New firms (prior year) | 16.64 | 0.184 | 0.74 | ++ | 0.89 | ++ | | | 0.78 | ++ | 0.23 |
| Population density | 13.31 | 0.134 | | | | | | | | | 0.10 |
| Bank deposits | 11.62 | 0.072 | | | | | | | | | 0.00 |
| Population | 9.34 | 0.146 | 0.07 | ++ | 0.01 | ++ | | | 0.04 | ++ | 0.47 |
| Total employment | 9.30 | 0.075 | 0.03 | – | | | | | | | 0.05 |
| Business density | 8.60 | 0.059 | | | | | | | | | 0.04 |
| Pop. Aged 25–44 (%) | 8.38 | 0.004 | | | | | | | 0.00 | – | |
| Year | 8.34 | 0.010 | | | | | | | | | |
| Banks | 7.99 | 0.048 | | | 0.00 | – | 0.014 | ++ | | | |
| Self employement (%) | 7.79 | 0.003 | | | | | | | | | |
| Ethnic diversity | 7.29 | 0.003 | | | | | | | 0.00 | ++ | 0.00 |
| Industry diversity | 7.26 | 0.024 | | | | | | | | | 0.01 |
| County GDP | 7.22 | 0.015 | 0.03 | ++ | | | | | | | 0.00 |
| Industry size: manufacturing | 6.80 | 0.014 | | | | | | | | | 0.00 |
| Pop. dollege education (%) | 6.79 | 0.004 | | | | | | | | | |
| Industry size: healthcare | 6.77 | 0.019 | | | | | | | 0.00 | – | 0.00 |
| Distance from metro area | 6.70 | 0.006 | | | | | | | | | |
| Oil play: none | 5.68 | 0.002 | | | | | | | | | |
| Protestant ethic | 5.64 | 0.002 | | | | | | | | | |
| Gas production | 5.64 | 0.004 | | | | | | | | | |
| Unemployment rate | 5.21 | 0.004 | | | | | | | 0.00 | – | |
| Industry size: extraction | 5.01 | 0.002 | | | | | | | | | |
| Oil production | 4.95 | 0.006 | 0.00 | – | | | | | 0.00 | – | |
| Income inequality | 4.61 | 0.003 | | | 0.00 | – | | | | | |
| Industry size: elder and childcare | 4.35 | 0.001 | | | 0.00 | ++ | | | | | |
| Industry size: recreational | 4.30 | 0.011 | | | | | | | | | |
| Land area | 4.22 | 0.008 | | | | | | | | | |
| Resilience | 4.19 | 0.003 | | | | | | | | | |
| Broadband access | 4.06 | 0.002 | | | | | | | | | |
| Pop. in poverty (%) | 4.02 | 0.002 | | | | | | | | | |
| Social capital | 3.88 | 0.004 | | | | | | | 0.00 | ++ | |
| Natural amenities score | 3.78 | 0.004 | | | | | | | | | |
| Pop. Aged 65 and older (%) | 3.69 | 0.003 | | | | | | | 0.00 | ++ | |
| Inmigration (%) | 3.69 | 0.005 | | | 0.00 | ++ | 0.004 | – | | | |
| Industry size: K-12 education | 3.19 | 0.001 | | | | | | | | | 0.00 |
| Outmigration | 3.06 | 0.004 | | | | | | | 0.00 | – | |
| Percent insured | 2.55 | 0.004 | | | | | | | | | |
| Patents | 2.09 | 0.004 | | | | | | | | | 0.01 |
| Federal funds rate[3] | 2.00 | 0.002 | 0.00 | – | | | | | | | |
| Wind capacity | 1.37 | 0.001 | | | | | | | 0.02 | ++ | |
| Failed firms (prior year) | 0.35 | 0.012 | | | 0.01 | ++ | | | | | 0.08 |
| Oil play: Eagle Ford | | | | | 0.00 | – | 0.003 | ++ | | | |

(*Continued*)

**Table 4.** (Continued)

| | RANDOM FOREST | | Multilinear regression[1] | | Forward subset | | Backward subset | | Lasso | | XG boost |
|---|---|---|---|---|---|---|---|---|---|---|---|
| Variable[2] | % inc MSE | Inc node purity | Coef. | +/- | Coef. | +/- | Coef. | +/- | Coef. | +/- | Gain |
| Oil play: Granite Wash | | | | | 0.00 | – | | | | | |

[1] Coefficients significant at 10% level. Individual year dummy variables not reported in this table

[2] For ease of viewing, we have removed those variables which contributed least to MSE and which appeared in no other models. In order of least importance, They are: Percent industry community college; percent industry coal; percent industry oil and gas; solar installations; percent industry farming; presidential vote differential; and percent of residents born in-county.

[3] Constant across all years in a given county or, in the case of Federal Funds Rate, across all counties each year. Broadband access is available for two years.

## Economic variables

Prior research has also identified strong associations between firm creation and economic variables, including access to capital [9], industry diversity [24], interest rates, changes in technology, and the rate of business failures [11]. From this perspective, demographics—education levels, unemployment rates, income equality—become measure of labor market readiness, and the availability of local workers has been shown a predictor of entrepreneurial outcomes [8]. We included a wide range of economic variables in our analysis; The most important across all models include the unemployment rate. Not salient were hypothesized economic factors such as total employment, county GDP, patents, interest rates, and business density.

Understood through the lens of labor market economics, the middling ranking total employment in our models stands in contrast to prior research, which suggests that low levels of unemployment make it less attractive for would-be entrepreneurs to start businesses and more difficult for them to hire employees [26]. More in line with previous research is the association between the number of local banks and firm creation. Internationally-focused research on firm creation across countries often includes variables regarding access to finance [10, 14], though US-focused research has not. This may be in part because banking in the United States has become so consolidated that the importance of *local* banking institutions with a physical presence has been overlooked.

Two surprising results are the relatively low predictive power of patents and business density on firm creation. Literature on patents suggests a strong set of links among patents, innovation and entrepreneurship [11]. But in two of our models, that relationship is significant and negative. This lines up with some recent scholarship suggesting that the United States is unusual in how much patents are concentrated in its large cities [68] and that patents are a less helpful proxy for innovation in rural areas than urban ones [69, 70]. Future work might examine this more closely by decomposing patents into invention, design, and plants patents. Business density, too, has been shown by prior research to be associated with economic resilience

**Table 5.** Comparison of 10-fold cross validation error.

| Model | Test MSE |
|---|---|
| Linear Regression | 0.000049 |
| Forward Subset Selection | 0.006819 |
| Backward Subset Selection | 0.006833 |
| Lasso Regression | 0.049521 |
| Random Forest | 0.000082 |
| XG Boost | 0.000085 |

**Table 6. Multi-model variable importance rankings, weighted by model performance.**

| Variable | Rank | Variable | Rank |
|---|---|---|---|
| Population | 1 | Pop. born in state (%) | 32 |
| New firms (prior year) | 2 | Pop. college education (%) | 33 |
| Oil production | 3 | Income inequality | 34 |
| Banks | 4 | Broadband access | 35 |
| Immigration (%) | 5 | Resilience | 36 |
| Industry size: healthcare | 6 | Land area | 37 |
| Industry size: elder and childcare | 7 | Distance from metro area | 38 |
| Failed firms (prior year) | 8 | Natural Amenities score | 39 |
| Wind capacity | 9 | Self-employment | 40 |
| Pop. aged 25–44 (%) | 10 | Patents | 41 |
| Pop. aged 65 and older (%) | 11 | Bank deposits | 42 |
| Ethnic diversity | 12 | Outmigration (%) | 43 |
| Unemployment rate | 13 | Population density | 44 |
| Total Employment | 14 | Year: 2010 | 45 |
| Social capital | 15 | Year: 2012 | 46 |
| County GDP | 16 | Industry size: oil and gas | 47 |
| Federal Funds Rate | 17 | Industry size: community college | 48 |
| Year: 2011 | 18 | Industry size: coal | 49 |
| Year: 2013 | 19 | Pop. in poverty (%) | 50 |
| Year: 2014 | 20 | Percent insured | 51 |
| Year: 2016 | 21 | Protestant Ethic | 52 |
| Industry size: farming | 22 | Year: 2009 | 53 |
| Industry size: extraction | 23 | Year: 2015 | 54 |
| Industry size: recreational | 24 | Oil play: Eagle Ford | 55 |
| Industry size: K-12 education | 25 | Year: 2017 | 56 |
| Industry size: manufacturing | 26 | Oil play: Granite Wash | 57 |
| Business density | 27 | (Intercept) | 58 |
| Industry diversity | 28 | Year: 2018 | 59 |
| Gas production | 29 | Oil play: Haynesville | 60 |
| Solar installations | 30 | Oil play: None | 61 |
| 2020 presidential vote differential (%) | 31 | Oil play: Permian | 62 |

[3]. But our results suggest that it is not greatly predictive of firm creation. Interpreting these two outcomes together, it may be that the agglomeration effects present in large cities may not be as powerful in urban areas, where more dedicated work is needed to induce spillover effects of innovation [71].

One variable receiving considerable attention across the country and in Texas specifically is access to broadband [21, 72]. While perhaps of importance in areas like equitable access to health care and education, broadband access does not emerge in this data as predictive of firm creation. We should note here that broadband access data was only available for two years; this is absence of evidence, not clear evidence of absence.

## Energy and other industry variables

Recent scholarship has explored the relationship between the oil and gas industry, using oil production both as an instrumental variable [73] and an explanatory one [26], finding that oil and gas expansion is negatively associated with firm creation. In our analysis, we re-examined this

relationship, along with two other energy industries (solar and wind) that have grown rapidly in Texas in the last twenty years, along with other sectors (manufacturing, farming, extraction, education, healthcare, and elderly care) that are of particular importance in many rural areas.

Of our industry variables, four emerged as most important: wind, oil, healthcare, and elder/childcare. The negative association between oil production and firm creation in two of our parametric models is not surprising, given prior research suggesting that large extractive industries depress firm creation [26]. It may be that firms that spring up in support of that industry are outnumbered by the firms crowded out by higher wages in the oil industry. That oil, but not gas, is negatively associated requires some explanation. One possibility is that extraction of oil and gas in Texas are more properly considered two industries than one, with different value chains—something that makes practical sense given the global integration of oil markets as compared to the primarily domestic use of natural gas in the United States.

The relationship between wind energy and firm creation is clear, novel, and worthy of further study, especially in comparison with the absence of a relationship between solar energy and firm creation. Prior research has examined the positive economic impacts of specific wind energy projects [74], but to our knowledge this is the first test of the association between wind energy and entrepreneurship.

### Comparing predictors of urban and rural entrepreneurship

In addition to the models above, which predict firm creation in rural counties, we ran a second set of models for non-rural counties. Table 7 compares these results. It includes the top fifteen

**Table 7. Comparing urban and rural predictors of firm creation.**

| Variable | Rural Ranking [a] | Urban Ranking |
|---|---|---|
| New firms (prior year) | 2 | 1 |
| Population | 1 | 9 |
| Wind capacity | 9 | 13 |
| Industry size: healthcare | 6 | 25 |
| Oil production | 3 | 20 |
| Pop. aged 25–44 (%) | 10 | 27 |
| Pop. aged 65 and older (%) | 11 | 44 |
| Ethnic diversity | 12 | 18 |
| Unemployment rate | 13 | 30 |
| Social capital | 15 | 46 |
| Banks | 4 | 38 |
| Immigration (%) | 5 | 26 |
| Failed firms (prior year) | 8 | 37 |
| Income inequality | 34 | 14 |
| Bank deposits | 42 | 3 |
| Total employment | 14 | 2 |
| Pop. college education (%) | 33 | 11 |
| 2020 presidential vote differential (%) | 31 | 24 |
| Gas production | 29 | 12 |
| Patents | 41 | 15 |
| Federal Funds Rate | 17 | 4 |
| Industry size: extraction | 23 | 17 |
| Industry size: elder and childcare | 7 | 56 |
| Oil play: None | 61 | 19 |

[a] Individual Year Variables Removed

predictive features in both rural and non-rural counties, according to our multiple-model variable importance rankings (holding aside the control variables for specific years). The list of factors that predict non-rural firm creation is materially different from those predicting non-rural entrepreneurship. Patents, local bank deposits, education levels, employment levels, and natural gas production emerged as influential predictors of urban firm creation. The first three of these, at least, are well-studied in existing (and mostly urban-focused) scholarship on firm creation but did not emerge as important predictors of firm creation in rural areas.

Predictive in rural areas, but less so in urban areas are the size of the healthcare and elder/childcare industries, oil production, age distribution, the unemployment rate, social capital, the number of local banks, and in-migration. Common to both are population size, the number of firms created in the previous year, wind capacity, and ethnic diversity.

## Discussion

These results contribute to an overlapping set of conversations among both academics and practitioners of public policy. First, they contribute to research on rural entrepreneurship, by identifying a set of variables that merit deeper study (e.g. the size of the healthcare and elder/childcare industries, age distribution, unemployment, social capital, ethnic diversity, immigration, the number of local banks, and the size of the wind and oil industries) and others that might not be as important as suggested by prior research primarily focused on entrepreneurship in urban areas (e.g. patents, education levels, and access to broadband). Second, they offer a roadmap for public policy scholars that might be interested in using machine learning in other policy disciplines. Third, they offer policymakers interested in rural entrepreneurship a different set of tools for initially approaching policy problems than do traditional econometric models. Fourth, in the process they suggest a different way for policymakers to interface with academics on complex policy problems where tightly identifying causal mechanisms may be elusive or even unnecessary.

In reading each of the sections below, it is important to consider how and whether the results from Texas are generalizable to another context. In some ways, Texas makes up a reasonable starting point for studying the United States as a whole, in that it makes up nearly a tenth of its population and economic output; many of the dynamics present in the broader American economy are represented or even driven by Texas. But Texas is unique in many ways, including that its economy grew throughout most of the Great Recession, thanks in large part due to its hydraulic fracturing boom, which occurred during the period under study.

What is more generalizable than the specific results are the methods we propose that researchers—and more importantly, policymakers—can use to identify where to focus their limited attention. Researchers and policymakers in other geographies will want to add other geographically unique parameters to these models, to test local hypotheses about what might contribute to economic growth in that area. In Texas, for example, boosters of solar panels might claim that installation of solar panels will lead to greater economic and job growth. That feature did not prove predictive here, but it might in another geography.

### Implications for rural entrepreneurship research

Existing research on rural entrepreneurship is epistemologically and empirically disjointed. That is not a surprise on an issue as complex as the creation of new firms. One logically sound approach to such a complex issue is to break it into smaller, model-able parts. This approach has led researchers to clarify on the causal relationship between some variables and firm creation. But it has also led to an econometric version of the parable of multiple blindfolded investigators attempting to describe the proverbial elephant. A full, clear description of a single

feature of an elephant may get us a little closer to describing the whole beast, but it is hard to know that we have isolated the most useful piece of information.

For research on entrepreneurial determinants, a machine learning-driven approach offers an avenue for us to take a step back from a complex problem and observe its multiple theoretical parts *empirically*, identifying which facets merit greater attention. In the case of rural entrepreneurship, those factors include the age distribution, ethnic diversity, migration patterns and social capital of a rural population; specific industries, such as oil, wind, healthcare, and local banking; and some economic variables, such as the unemployment rate. It may also reveal which features might have gotten too much attention, either because of political palatability (such as access to broadband) or because existing research, focused on urban areas, focuses on factors (such as patents) that fail to translate into rural contexts.

## Implications for rural entrepreneurship policy

These results can help policymakers in two ways. First, they can help policymakers base decisions on better predictions of what kind of firm creation to expect in their rural counties and states. Second, they can help policymakers identify what factors should get more or less time on the research and policy agenda. In many of these cases, a direct policy may not be possible —a county cannot grow its population by fiat, for example. But it can work together, as some small towns across the U.S. have, to create enough linkages with other nearby areas to operate as a bloc, perhaps mimicking some of the advantages of larger populations.

In other areas, this research suggests other potential levers that rural policymakers might want to examine more closely, including the number of local banks; attracting younger and more racially diverse workforces; investing more in healthcare infrastructure; and (where geographically feasible) exploring wind energy production, for example. It should be reiterated here that this research establishes predictive relationships, not causal ones: It can help us prioritize which policies to investigate more deeply and which to deprioritize. However, estimating the potential effects of a particular policy option requires a shift from examining prediction to establishing causation.

In other cases, these results may suggest levers that policymakers cannot pull. Counties in Southeast Texas, for example, might not be able to develop wind capacity. But there may be other major infrastructure projects whose contracting could be made to look more like those used in the wind industry, potentially creating conditions for firm creation.

The results also suggest that the importance of some pro-entrepreneurship policies proposed by lobbying groups and think tanks, such as greater access to broadband, is not clear when ranked against other predictive variables. And these results suggest that some well-studied variables shown to be important in urban areas, such as the number of patents, are less predictive of firm growth in rural areas. If the presence of variables provides a clue as to what policymakers should prioritize, the repeated absence of variables suggests variables they should consider deprioritizing.

Some of the variables that emerged in these models as important are just as complex as those of firm creation. How would a rural county increase the number of local banks, for example? This question, too, is surrounded by a fog of often-disconnected policy briefs and consultant recommendations; getting a clear answer of where to focus lends itself equally well to the type of multi-method machine learning we have conducted in this paper.

## Implications for public policy scholarship

The process laid out here offers a model for how policy scholars might approach other kinds of questions, under certain conditions. First, these methods are most helpful when scholars are

concerned with comparing policy options rather than identifying the impact of a particular intervention. If an independent variable of interest has been identified, traditional regression methods may be more appropriate. But when dealing with policy problems as complex yet nascent as rural entrepreneurship, scholars often apply multiple theoretical approaches to a particular problem and seem to talk past each other. In such an example, the initial research goal may be to establish the *relative* importance of the independent variable to others.

Second, these methods are most useful under certain data conditions: Specifically, with large data sets, and those where the number of predictors ($p$) is much larger than the number of observations ($n$). Over the last couple of decades, many machine learning methods have been designed precisely to overcome this curse of dimensionality, which is expressed in shorthand as $p>>n$. Conversely, where there are a great many more observations than predictors than observations ($n>>p$), classical econometric models may be more appropriate depending on the nature of the data and may provide the necessary leverage for pointing to causal mechanisms. For $p>n$ and even $n>p$ (as in this dataset), there is less clear-cut guidance on the types of models to use. The same is true for complex datasets with many features (independent variables) in applied domains, such as rural entrepreneurship, that are still evolving and so do not have a firmly established set of features that are standard across all models. Under these circumstances, the analytical framework we present in this paper could serve well to advance understanding of the underlying data and to offer fruitful pointers for deeper study.

### Lessons for policymakers interested in complex problems

For decades, policymakers looking to implement evidence-based policy have turned to research for answers. That research has turned most often to a set of linear regression models designed to identify as tightly as possible the causal relationship between two variables, often transformed into just the right shape for analysis. Even when those results are meaningful, they can only answer one type of question that policymakers might ask. Machine learning techniques offer a new and growing set of tools to help policymakers answer additional questions about what policy inputs they should prioritize given a particular policy outcome of interest. Answering such questions is especially important in the early phases during which the policy agenda is still taking shape before fully crystallizing. In instances where policymakers' task is to identify where to put scarce public resources to develop deeper and more robust understanding of potential solutions, or in scenario planning situations where prediction is the central task, predictive machine learning methods can prove useful additional tools to traditional causality-seeking approaches.

These models are not silver bullets, for at least two reasons. First, the added complexity of the models means that they are open to misinterpretation or intentional manipulation (or what might be called *disinterpretation*). [28, 40] provide two recent discussions of these issues with the application of machine learning in public policy research. Second, machine learning models are just a tool, and when wielded without care have the potential to be discriminatory. In the human context, data is one manifestation of reality, created in a particular socio-political context, and as such is often shrouded with complex, idiosyncratic drivers. Who gets to interpret the data and models is inextricable from questions of power in that context. [75, 76] offer discussions of these dangers and how they might be addressed.

This is not a new problem, however, nor one unique to the kind of computational methods we use in this paper. In fact, the potential for policy actors to abuse emerging methods was one inspiration for the multi-method approach we lay out here. Data is always collected by and filtered through the hands of humans embedded in institutions with their own biases. This multi-method approach has the potential to mitigate that in two ways: first, it provides a

mechanism through which more data can be incorporated into analysis, lowering the possibility of omitted variable bias. Second, it makes it harder for a policy actor to choose a single model that supports their agenda. Third, it has the potential to spur cross-agency coordination on data collection, and especially for the variables that show up as important in limited variable analyses and our larger variable pool study. With these dangers and possibilities in mind, we interpret our and others' models humbly—and we encourage our readers to do the same, especially when our results line up with our prior beliefs and biases.

## Supporting information

**S1 Appendix.** Appendix A: Descriptive Statistics of All Variables. Appendix B: Variable appearance across all models. Appendix C: Hyperparameter tuning plots.
(DOCX)

**S1 File.**
(ZIP)

## Acknowledgments

We thank UT Austin's IC2 Institute, and Art Markman and Gregory Pogue in particular, for their support for this research; Melinda Taylor, for her early encouragement and guidance; Paul von Hippel, Daniel Armanios and Tim Fitzgerald for their feedback; Megan Morris for being a supportive colleague throughout; and Matt Worthington and Fritz Boettner for their generosity with their data.

## Author Contributions

**Conceptualization:** Mark C. Hand, Vivek Shastry, Varun Rai.

**Data curation:** Mark C. Hand, Vivek Shastry.

**Formal analysis:** Mark C. Hand, Vivek Shastry.

**Funding acquisition:** Mark C. Hand, Varun Rai.

**Investigation:** Mark C. Hand, Vivek Shastry.

**Methodology:** Mark C. Hand, Vivek Shastry, Varun Rai.

**Project administration:** Mark C. Hand, Vivek Shastry, Varun Rai.

**Resources:** Vivek Shastry, Varun Rai.

**Supervision:** Varun Rai.

**Validation:** Mark C. Hand, Vivek Shastry, Varun Rai.

**Visualization:** Mark C. Hand, Vivek Shastry.

**Writing – original draft:** Mark C. Hand, Vivek Shastry.

**Writing – review & editing:** Mark C. Hand, Vivek Shastry, Varun Rai.

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
