## [Decision Letter · Decision Letter 0]

18 Oct 2022

PONE-D-22-12529Predicting Firm Creation in Rural Texas: A Multi-Model Machine Learning Approach to a Complex Policy ProblemPLOS ONE

Dear Dr. Hand,

Thank you for submitting your manuscript to PLOS ONE. After careful consideration, we feel that it has merit but does not fully meet PLOS ONE’s publication criteria as it currently stands. Therefore, we invite you to submit a revised version of the manuscript that addresses the points raised during the review process.

I recommend that it should be revised taking into account the changes requested by the reviewers. Since the requested changes include valuable and constructive reviews, I would like to give you a chance to revise your manuscript. The revised manuscript will undergo the next round of review by two reviewers.

We look forward to receiving your revised manuscript.

Kind regards,

Baogui Xin, Ph.D.

Academic Editor

PLOS ONE

2.Thank you for stating the following in the Acknowledgments Section of your manuscript:

“We thank UT Austin’s IC2 Institute, and Art Markman and Gregory Pogue in particular, for supporting this research; Melinda Taylor, for her early support and encouragement; Paul von Hippel, Daniel Armanios and Tim Fitzgerald for their early feedback; Megan Morris for her thoughtful support throughout; and Fritz Boettner for generosity with his data.”

“This study was funded by a grant to VR by the IC2 Institute at the University of Texas at Austin (https://ic2.utexas.edu/). The funders had no role in study design, data collection and analysis, decision to publish, or preparation of the manuscript.”

Reviewers' comments:

Reviewer's Responses to Questions

**Comments to the Author**

1. Is the manuscript technically sound, and do the data support the conclusions?

Reviewer #1: Partly

Reviewer #2: Partly

2. Has the statistical analysis been performed appropriately and rigorously? 

Reviewer #1: No

Reviewer #2: I Don't Know

3. Have the authors made all data underlying the findings in their manuscript fully available?

Reviewer #1: No

Reviewer #2: Yes

4. Is the manuscript presented in an intelligible fashion and written in standard English?

Reviewer #1: Yes

Reviewer #2: Yes

5. Review Comments to the Author

Reviewer #1: REVIEW

“Predicting Firm Creation in Rural Texas: A Multi-Model Machine Learning Approach to a Complex Policy Problem”

PONE-D-22-12529

Plos One

This paper examines the factors driving the creation of firms in rural America. In doing so, the authors employ a Multi-Model Machine Learning Approach. The authors find that some factors that promote entrepreneurship may not be as predictive as socioeconomic ones. Moreover, the strength of specific industries predicts firm growth, as does the number of local banks. Finally, the authors provide some policy implications of their findings.

COMMENTS:

The paper examines a very interesting and policy-relevant research question. In this sense, the paper's findings are interesting but there are some points that the authors should address before considering publishing the paper at Plos One.

Let me summarize my comments and suggestions below:

1) Texas: External validity

The authors should explain better why Texas could be a good laboratory to conduct this analysis. Are there any features that make Texas different from other states (such as Vermont or Indiana). This point is relevant for the external validity of the analysis. If Texas is largely different than other rural states it could be the case that the specificities of Texas are the ones driving the results. I am not suggesting that Texas is not a good laboratory just the need to explain and justify it.

2) Hyper-parameters:

Given that the hyper-parameters (e.g. number of trees, number of features in each tree, etc.) used by the random forest are arbitrarily selected, it is important to discuss why no optimization took place and which criteria were used exactly for making any choices. These hyper-parameters should be reported as it is standard in the literature (Albanesi & Vamossy, 2019; Kou et al., 2021; Obrizan et al., 2019; Petropoulos et al., 2019). Also, it would be nice to perform a sensitivity analysis, reporting how the importance of the variables changes when other hyper-parameters are being used. Are the results robust, or do they depend on the parameters and initializations considered?. In the current form of the manuscript, the operations performed are not reproducible.

3) Other machine learning methods.

The authors should justify the machine learning techniques used. Why did the authors choose the random forest algorithm rather than other classification techniques (e.g. Extreme Gradient Boosting). This technique (Extreme Gradient Boosting) is also commonly used in these studies and it also provides a ranking of the variable in terms of importance (Carbo-Valverde et al., 2020; Carmona et al., 2019; Fuster et al., 2018; Obrizan et al., 2019). This point should be discussed in the paper.

4) Ranking the variables: Random Forest

The random forest algorithm allows the authors to rank the variables based on 1) the mean decrease in accuracy (which reflects the mean loss in accuracy when each specific variable is excluded from the regression algorithm) and 2) (the mean decrease in Gini (which reflects how each feature contributes to the homogeneity between the decision trees used in the resulting random forest). However, the authors rank the variables just based on the mean decrease in accuracy (“In the random forest model, we rank predictors according to their contribution to reducing the percent mean-squared error.”). I think it could be interesting to consider also the mean decrease in Gini. In this line, the authors could follow the approach employed by Carbo-Valverde et al., (2020). This paper 1) ranks every variable using the mean decrease in accuracy and the mean decrease in Gini, respectively, 2) scores each variable, 3) computes the total score of each variable, and 4) reorders the variables by the total score. This approach would allow the authors to provide a rank combining both measures (Accuracy and Gini).

5) Data description

The paper does not provide any summary statistics of the main variables employed in the analysis. The paper does not report the number of companies created in rural counties of Texas over time. The authors should provide some summary statistics (mean, min, max, median, sd) of the main variables used.

6) Sample period

The authors examine the creation of firms in rural Texas from 2008 to 2018. However, during this large period, there are different phases. From 2008 to 2012, the whole country was facing the Global Financial Crisis (GFC), one of the greatest financial crises. I would expect a relatively low number of firms created during these years. However, from 2012 to 2018, the economy was recovering, so the number of firms created in those years will be relatively higher compared to those created during the GFC. I think that it would make sense to split the sample period into different phases (at least as a robustness check)?

Minor comments:

• Footnote 2 (page 15) is missing.

References

- Albanesi, S., & Vamossy, D. F. (2019). Predicting Consumer Default: A Deep Learning Approach. NBER Working Paper Series, N. 26165.

- Carbo-Valverde, S., Cuadros-Solas, P., & Rodríguez-Fernández, F. (2020). A machine learning approach to the digitalization of bank customers: Evidence from random and causal forests. In PLoS ONE (Vol. 15, Issue 10 October). https://doi.org/10.1371/journal.pone.0240362

- Carmona, P., Climent, F., & Momparler, A. (2019). Predicting failure in the U . S . banking sector : An extreme gradient boosting approach. International Review of Economics and Finance, 61(March 2018), 304–323. https://doi.org/10.1016/j.iref.2018.03.008

- Fuster, A., Goldsmith-Pinkham, P., Ramadorai, T., & Walther, A. (2018). Predictably Unequal ? The Effects of Machine Learning on Credit Markets. Review of Financial Studies, Forthcomin(November).

- Kou, G., Xu, Y., Peng, Y., Shen, F., Chen, Y., Chang, K., & Kou, S. (2021). Bankruptcy prediction for SMEs using transactional data and two-stage multiobjective feature selection. Decision Support Systems, 140, 113429. https://doi.org/10.1016/j.dss.2020.113429

- Obrizan, M., Torosyan, K., & Pignatti, N. (2019). Tobacco Spending in Georgia: Machine Learning Approach. In Y. Kondratenko, G. Kondratenko, & I. Sidenko (Eds.), Recent Developments in Data Science and Intelligent Analysis of Information (Vol. 836, pp. 71–80). Springer International Publishing. https://doi.org/10.1007/978-3-319-97885-7

- Petropoulos, A., Siakoulis, V., Stavroulakis, E., & Klamargias, A. (2019). A robust machine learning approach for credit risk analysis of large loan level datasets using deep learning and extreme gradient boosting. BIS.

Reviewer #2: The article uses a variety of machine learning methods to predict business growth in rural counties in Texas, providing policymakers with a set of tools different from traditional econometric models for initial policy solutions. The research methodology and process are interesting and challenging, but again, some areas are questionable.

(1) Machine learning approaches are generally not built on interpretability, and there is insufficient evidence in the paper to demonstrate that the accuracy of predictions is due to good luck or the validity of the method. If the authors can futher strengthen the externalities of the study findings, such as using other region’s data for prediction and comparison of accuracy with Texas region, it will be more persuasive.

(2) Why authors applied linear regression, forward and backward subset selection of linear regression, lasso regression, and random forest among hundreds ways? As I know, they are not considered as novel algorithms or SOTA model nowdays. They are maybe more suitable for this particular problem, but the selection reasonability of above models need to be futher justified by involving context or literatures in the paper. I’m also confused about "the weighted ranking of all models is combined, their mean values are taken, and the combined measures are ranked again". What is the basis for taking the mean weighted ranking? It is necessary to explain more clearly about the selection and combination of sub-models in the multi-model interpretation framework.

(3) It’s not rigorous enough about the explanation for the relatively low predictive power of patents and/or business density (patenting efforts are likely to be concentrated in modern technology-driven industries, which are overwhelmingly located in large cities, while in rural areas, having large, dominant patent-producing firms can discourage the establishment of other potential firms), because most empirical studies have proved there is positive effect by knowledge diffusion. Even if the author's judgment is correct, it’s more reasonable to break down patents into inventions, designs, and plant patents instead of using the total number of patents to prove it.

(4) The factors that affect the creation of enterprises, in reality, are complex, and machine learning through passively observed data should ensure the accuracy of the data. In the article, more variables are used, and whether the measurement of variables from different sources will bring errors and thus affect the reliability of the final prediction results. The multi-model explanatory model may rely to some extent on parameter adjustment and function selection, and the final results may be more sensitive to missing values, and it is questionable how to ensure the stability of the model prediction.

6. PLOS authors have the option to publish the peer review history of their article (what does this mean?). If published, this will include your full peer review and any attached files.

Reviewer #1: No

Reviewer #2: No

---

## [Author Response · Author response to Decision Letter 0]

31 Mar 2023

Reviewer #1

This paper examines the factors driving the creation of firms in rural America. In doing so, the authors employ a Multi-Model Machine Learning Approach. The authors find that some factors that promote entrepreneurship may not be as predictive as socioeconomic ones. Moreover, the strength of specific industries predicts firm growth, as does the number of local banks. Finally, the authors provide some policy implications of their findings.

The paper examines a very interesting and policy-relevant research question. In this sense, the paper's findings are interesting but there are some points that the authors should address before considering publishing the paper at Plos One.

Let me summarize my comments and suggestions below:

1) Texas: External validity

The authors should explain better why Texas could be a good laboratory to conduct this analysis. Are there any features that make Texas different from other states (such as Vermont or Indiana). This point is relevant for the external validity of the analysis. If Texas is largely different than other rural states it could be the case that the specificities of Texas are the ones driving the results. I am not suggesting that Texas is not a good laboratory just the need to explain and justify it.

> First, we’d like to say thank you to Reviewer #1 for your review. We appreciate your engaging meaningfully with both the methods and how we talk about the results, and we have addressed each of your concerns in the paper, as described below. 

> On the question of geography, we have added a mention of this in introducing our data (p9) and a paragraph in the discussion (pp. 25-27), explaining how our results here may or may not be generalizable. In some ways, we believe, rural Texas is large and diverse enough for our results to serve as a reasonable starting point for studying other geographies. In other ways—e.g., the fracking boom in Texas during the period under study—it is not; each state will have its own unique economic mix. Methodologically, we argue that the process we went through in this paper is generalizable, even if researchers focused on other geographies would want to add other economic parameters unique to that state. 

2) Hyper-parameters:

Given that the hyper-parameters (e.g. number of trees, number of features in each tree, etc.) used by the random forest are arbitrarily selected, it is important to discuss why no optimization took place and which criteria were used exactly for making any choices. These hyper-parameters should be reported as it is standard in the literature (Albanesi & Vamossy, 2019; Kou et al., 2021; Obrizan et al., 2019; Petropoulos et al., 2019). 

> We thank the reviewer for bringing this to our notice. We have indeed optimized the hyperparameters wherever appropriate. We have included the additional explanation in the revised manuscript (p16), and included the associated figures in the appendix. The random forest model was tuned to select the optimal mtry value (where mtry is the number of variables randomly sampled at each split). In the lasso regression model, cross validation was used to select the lambda value which minimizes the mean cross-validated error. In the XG boost model, a grid search with 5-fold cross validation was performed to tune five hyperparameters - the maximum depth of a tree (max_depth), subsample ratio of the training instance (subsample), subsample ratio of columns when constructing each tree (colsample_bytree), lambda, and alpha values. 

Also, it would be nice to perform a sensitivity analysis, reporting how the importance of the variables changes when other hyper-parameters are being used. Are the results robust, or do they depend on the parameters and initializations considered?. 

> As mentioned above, wherever appropriate we tuned the hyperparameters to minimize the cross-validation errors. Since this optimization was already performed, we did not conduct any additional sensitivity analysis for the hyperparameters. 

In the current form of the manuscript, the operations performed are not reproducible.

> We noticed that you responded “No” to “have the authors made all data underlying the findings in their manuscript fully available?” We have re-attached our updated analysis here, in addition to uploading it to protocols.io per the editor’s suggestions. We have made this clear in the blind version of the text in a footnote on p9. 

3) Other machine learning methods.

The authors should justify the machine learning techniques used. Why did the authors choose the random forest algorithm rather than other classification techniques (e.g. Extreme Gradient Boosting). This technique (Extreme Gradient Boosting) is also commonly used in these studies and it also provides a ranking of the variable in terms of importance (Carbo-Valverde et al., 2020; Carmona et al., 2019; Fuster et al., 2018; Obrizan et al., 2019). This point should be discussed in the paper.

> Thank you. We took two steps in response to this comment. First, we added back in text (p15) that had been in a footnote and was somehow omitted in this draft, explaining why we chose the methods we did. Second, we have added extreme gradient boosting (see pp 15-16), at your suggestion. When we did this, our final outputs shifted: Year variables dropped in combined importance, for example. County GDP and the federal funds rate rose in importance, while outmigration fell. None of the other changes altered our interpretation of our results, though the changes do, we believe, underscore our argument that using multiple methods can offset each individual method’s idosyncracies. 

4) Ranking the variables: Random Forest

The random forest algorithm allows the authors to rank the variables based on 1) the mean decrease in accuracy (which reflects the mean loss in accuracy when each specific variable is excluded from the regression algorithm) and 2) (the mean decrease in Gini (which reflects how each feature contributes to the homogeneity between the decision trees used in the resulting random forest). However, the authors rank the variables just based on the mean decrease in accuracy (“In the random forest model, we rank predictors according to their contribution to reducing the percent mean-squared error.”). I think it could be interesting to consider also the mean decrease in Gini. In this line, the authors could follow the approach employed by Carbo-Valverde et al., (2020). This paper 1) ranks every variable using the mean decrease in accuracy and the mean decrease in Gini, respectively, 2) scores each variable, 3) computes the total score of each variable, and 4) reorders the variables by the total score. This approach would allow the authors to provide a rank combining both measures (Accuracy and Gini).

> We thank the reviewer for this suggestion. As per the suggestion, we implemented the ranking by decrease in Gini as well using the ranking by increase in node purity (IncNodePurity) option in the random forest function. Since we have a continuous dependent variable, the increase in node purity was calculated in terms of RSS. We have included this list of variables (“Random Forest RSS”) in the calculation of the final ranking. We have updated the text on page 16 and tables on p21 accordingly; see our response above for how our results changed once we added in this and XG Boost. 

5) Data description

The paper does not provide any summary statistics of the main variables employed in the analysis. The paper does not report the number of companies created in rural counties of Texas over time. The authors should provide some summary statistics (mean, min, max, median, sd) of the main variables used.

> On p9 we have added a figure showing rural firm creation all years in our data set. In an appendix, we have included a similar table for all our variables, a table we felt unwieldy for the body of the paper. 

6) Sample period

The authors examine the creation of firms in rural Texas from 2008 to 2018. However, during this large period, there are different phases. From 2008 to 2012, the whole country was facing the Global Financial Crisis (GFC), one of the greatest financial crises. I would expect a relatively low number of firms created during these years. However, from 2012 to 2018, the economy was recovering, so the number of firms created in those years will be relatively higher compared to those created during the GFC. I think that it would make sense to split the sample period into different phases (at least as a robustness check)?

> We agree with the reviewer’s observation. We re-examined the trend in new firm creation from 2008 to 2018, and we did not notice any abrupt changes in the rate of firm creation pre and post 2012, but rather a consistent increase in firm starts beginning in 2010. We therefore did not run separate models for pre and post 2012. And since our objective was to use historical data to predict the current rates of firm creation, we concluded that such a split was not appropriate in our context. However, we have incorporated year dummies in our model as a way of controlling for any variation over time, and in fact many year variables (2011, 2013, 2014, 2016) emerge as significant predictors. 

Minor comments:

• Footnote 2 (page 15) is missing.

> Thank you! Removed (see above). And thank you for including your list of helpful references (omitted here for brevity), all of which are now included in the manuscript. 

Reviewer #2

The article uses a variety of machine learning methods to predict business growth in rural counties in Texas, providing policymakers with a set of tools different from traditional econometric models for initial policy solutions. The research methodology and process are interesting and challenging, but again, some areas are questionable.

(1) Machine learning approaches are generally not built on interpretability, and there is insufficient evidence in the paper to demonstrate that the accuracy of predictions is due to good luck or the validity of the method. 

> Thank you for engaging meaningfully with our manuscript, and for your helpful comments. This comment points in the same direction as our overall argument: each individual machine learning method performs differently on the same data set, so using multiple methods is a more policy-relevant way of analyzing data. For each individual model, though, we use standard tuning (optimization) to minimize errors—or put another way, to take as much good luck out of the equation as possible. 

If the authors can futher strengthen the externalities of the study findings, such as using other region’s data for prediction and comparison of accuracy with Texas region, it will be more persuasive. 

> On the question of geography, we have added a mention of this in introducing our data (p9) and a paragraph in the discussion (pp 25-27), explaining how our results here may or may not be generalizable. In some ways, we believe, rural Texas is large and diverse enough for our results to serve as a reasonable starting point for studying other geographies. In other ways—e.g., the fracking boom in Texas during the period under study—it is not; each state will have its own unique economic mix. Methodologically, we argue that the process we went through in this paper is generalizable, even if researchers focused on other geographies would want to add other economic parameters unique to that state. We hope that the reviewer will understand that the process of collecting state-specific data for other states would require building relationships with state-level agencies, as we did here for county-level oil and gas data. We hope you will agree that is outside the scope of this paper, though a fruitful avenue for future work! 

(2) Why authors applied linear regression, forward and backward subset selection of linear regression, lasso regression, and random forest among hundreds ways? As I know, they are not considered as novel algorithms or SOTA model nowdays. They are maybe more suitable for this particular problem, but the selection reasonability of above models need to be futher justified by involving context or literatures in the paper. 

> Thank you. First, we added back in text (p15) that had been in a footnote we accidentally omitted in the previous draft, explaining why we chose the methods we did. First, we chose the subset of machine learning methods with outputs that include some rank of variable importance that is comparable across methods. That is now in the main text on p15. Second, and related, the outputs of these methods are relatively easy to explain to policymakers unfamiliar with machine learning methods, unlike some of the more state of the art methods. In response to your comment and at Reviewer 1’s suggestion, however, we have also added extreme gradient boosting, which fits both criteria (see pp 19-20). 

I’m also confused about "the weighted ranking of all models is combined, their mean values are taken, and the combined measures are ranked again". What is the basis for taking the mean weighted ranking? It is necessary to explain more clearly about the selection and combination of sub-models in the multi-model interpretation framework.

> On the selection of sub-models, we hope we have addressed in responding to your question above (and on p15 of the text). The optimization strategies we deployed leaves us with a single maximally predictive model for each algorithm. 

> As for combination, we could not find this exact quotation in our manuscript but we have edited the section A Multi-Model Interpretation Framework (now p17) to be clearer. Here is the new text: 

> The combined interpretation framework consists of three steps (Fig 3). First, we take the list of variables from each model output and rank them according to their importance. The ranking scheme depends on the nature of output from each model. In the random forest model, we rank predictors according to their contribution to reducing the percent mean-squared error. In all other models, we rank the predictors according to the magnitude of their (standardized) coefficients. This ensures that the predictors that are more important in each model get higher ranks. Further, the predictors missing in each model and predictors that are not statistically significant in the linear regression model are penalized by assigning their rank as one greater than the total variables in the largest model. This ensures that the predictors that are not significant in each model get lower ranks. 

> Second, once the rank for each predictor is assigned for all models, these ranks are weighted by the respective model’s predictive accuracy. In our case for example, we weight the ranks from each model by that model’s 10-fold cross validation error. This ensures that the predictors in better performing models get higher weightage. 

> Finally, we combine the weighted ranks from all models. To do this, we take the average rank of each predictor across all models. Then, we rank those averages. This ensures that the predictors appearing across multiple models get higher ranks. This is the final measure of each predictor’s overall importance in our models, as described below.

(3) It’s not rigorous enough about the explanation for the relatively low predictive power of patents and/or business density (patenting efforts are likely to be concentrated in modern technology-driven industries, which are overwhelmingly located in large cities, while in rural areas, having large, dominant patent-producing firms can discourage the establishment of other potential firms), because most empirical studies have proved there is positive effect by knowledge diffusion. Even if the author's judgment is correct, it’s more reasonable to break down patents into inventions, designs, and plant patents instead of using the total number of patents to prove it.

> This feedback was helpful in pushing us to examine our interpretation of these results. What we found and added to the paper (see p22) was prior work questioning the applicability of existing patent measures and theories about agglomeration effects to rural areas. We also added as potential future work the reviewer’s suggestion that patents might be split out according to patent type, something we were unable to accomplish within the scope of this paper. 

(4) The factors that affect the creation of enterprises, in reality, are complex, and machine learning through passively observed data should ensure the accuracy of the data. In the article, more variables are used, and whether the measurement of variables from different sources will bring errors and thus affect the reliability of the final prediction results. The multi-model explanatory model may rely to some extent on parameter adjustment and function selection, and the final results may be more sensitive to missing values, and it is questionable how to ensure the stability of the model prediction. 

> We agree with the reviewer’s points about the complexity of the question of firm creation, and the dangers of relying on passively collected data. In fact, those observations are the motive behind the deployment of multiple methods, each with its advantages and weaknesses, rolled up into a simple and transparent manner that policymakers can quickly understand and then push back against. 

> To address the reviewer’s specific question about parameter adjustment (echoed by Reviewer 1) we have:

> • made clearer that we optimized hyper-parameters where possible (see p16 and Appendix C), to reduce the kind of reliance on parameter adjustment the reviewer identifies as a potential issue

> • added a new paragraph to our conclusion, laying out our perspective on those questions. The final two paragraphs now read as follows: 

> These models are not silver bullets, for at least two reasons. First, the added complexity of the models means that they are open to misinterpretation or intentional manipulation (or what might be called disinterpretation). [28] and [38] provide two recent discussions of these issues with the application of machine learning in public policy research. Second, machine learning models are just a tool, and when wielded without care have the potential to be discriminatory. In the human context, data is one manifestation of reality, created in a particular socio-political context, and as such is often shrouded with complex, idiosyncratic drivers. Who gets to interpret the data and models is inextricable from questions of power in that context. [73] and [74] offer discussions of these dangers and how they might be addressed. 

> This is not a new problem, however, nor one unique to the kind of computational methods we use in this paper. In fact, the potential for policy actors to abuse emerging methods was one inspiration for the multi-method approach we lay out here. Data is always collected by and filtered through the hands of humans embedded in institutions with their own biases. This multi-method approach has the potential to mitigate that in two ways: first, it provides a mechanism through which more data can be incorporated into analysis, lowering the possibility of omitted variable bias. Second, it makes it harder for a policy actor to choose a single model that supports their agenda. Third, it has the potential to spur cross-agency coordination on data collection, and especially for the variables that show up as important in limited variable analyses and our larger variable pool study. With these dangers and possibilities in mind, we interpret our and others’ models humbly—and we encourage our readers to do the same, especially when our results line up with our prior beliefs and biases.

> Lastly, thank you again for your detailed engagement with our manuscript, which we believe is stronger for it.

---

## [Decision Letter · Decision Letter 1]

4 May 2023

PONE-D-22-12529R1Predicting Firm Creation in Rural Texas: A Multi-Model Machine Learning Approach to a Complex Policy ProblemPLOS ONE

Dear Dr. Hand,

Thank you for submitting your manuscript to PLOS ONE. After careful consideration, we feel that it has merit but does not fully meet PLOS ONE’s publication criteria as it currently stands. Therefore, we invite you to submit a revised version of the manuscript that addresses the points raised during the review process.

We recommend that it should be revised taking into account the changes requested by the reviewers. Since the requested changes includes Minor Revision, the revised manuscript will undergo the next round of review by the same reviewers or only by the Academic Editor.

We look forward to receiving your revised manuscript.

Kind regards,

Baogui Xin, Ph.D.

Academic Editor

PLOS ONE

Journal Requirements:

Reviewers' comments:

Reviewer's Responses to Questions

**Comments to the Author**

1. If the authors have adequately addressed your comments raised in a previous round of review and you feel that this manuscript is now acceptable for publication, you may indicate that here to bypass the “Comments to the Author” section, enter your conflict of interest statement in the “Confidential to Editor” section, and submit your "Accept" recommendation.

Reviewer #1: All comments have been addressed

Reviewer #2: (No Response)

2. Is the manuscript technically sound, and do the data support the conclusions?

Reviewer #1: Yes

Reviewer #2: Yes

3. Has the statistical analysis been performed appropriately and rigorously? 

Reviewer #1: Yes

Reviewer #2: I Don't Know

4. Have the authors made all data underlying the findings in their manuscript fully available?

Reviewer #1: Yes

Reviewer #2: Yes

5. Is the manuscript presented in an intelligible fashion and written in standard English?

Reviewer #1: Yes

Reviewer #2: Yes

6. Review Comments to the Author

Reviewer #1: REVIEW

“Predicting Firm Creation in Rural Texas: A Multi-Model Machine Learning Approach to a Complex Policy Problem”

PONE-D-22-12529

Plos One

This paper examines the factors driving the creation of firms in rural America. In doing so, the authors employ a Multi-Model Machine Learning Approach. The authors find that some factors that promote entrepreneurship may not be as predictive as socioeconomic ones. Moreover, the strength of specific industries predicts firm growth, as does the number of local banks. Finally, the authors provide some policy implications of their findings.

COMMENTS:

The authors have addressed all the comments that I raised before. They have done a great effort to include new machine learning methods. I believe the paper could be published in its current form.

Reviewer #2: The main problem is the introduction part, of course which can be written in a variety of ways, but some important things still need to be clarified:

1) Research objectives. The paper asks a general question about "what can predict firm creation in rural America". But obviously the first paragraph does not address around the essence of this question (what's legitimacy of your research objective? from which aspect you will cut into this issue? what you will get and why it's important in theory or practice?... ). Promoting rural entrepreneurship is certainly important, but does it have anything to do with the question you raised in the first sentence? If so, what is the logic? If your logic is that the US government can intervene to promote rural entrepreneurship based on what machine learning predicts, that please gives us enough evidences that this logic can be implemented at the state level of the United States... Don't leave anything behind which may make reviewers and audiance confused and to guess.

2) Research gap. Not recommend to demonstrate the existing research by table here, it's not enough to just throw out some evidences, stating logic and viewpoint is your job and not homework of audience. In the introduction part, our concern have not reached the specific determinants and varialbles level yet , but more concerned your comment about the progress of research at the general level. This will show your familiarity with the field, your understanding of the problem, and the legitimacy of your following research gap summary. Furthermore, your summary of the research gap is not logical enough to connect the research objectives and existing research.

3) Margin contribution. Obviously I didn't find it in introduction part. Explain what you have done to fill the research gap and what is the incremental academic value of doing so.

Generally speaking, the narrative in the introduction part is insufficient and appropriate, which hinders the audience's jugement of the importance and innovation about this research. Don't let reviewers and audiance to fill the blanc of (why, what, how and whether good you have done) for you, they don't like it.

7. PLOS authors have the option to publish the peer review history of their article (what does this mean?). If published, this will include your full peer review and any attached files.

Reviewer #1: No

Reviewer #2: No

---

## [Author Response · Author response to Decision Letter 1]

27 May 2023

With gratitude, please see the attached Response to Reviewers.docx.

---

## [Editor Report · Decision Letter 2]

2 Jun 2023

Predicting Firm Creation in Rural Texas: A Multi-Model Machine Learning Approach to a Complex Policy Problem

PONE-D-22-12529R2

Dear Dr. Hand,

We’re pleased to inform you that your manuscript has been judged scientifically suitable for publication and will be formally accepted for publication once it meets all outstanding technical requirements.

Kind regards,

Baogui Xin, Ph.D.

Academic Editor

PLOS ONE
---

## [Editor Report · Acceptance letter]

7 Jun 2023

PONE-D-22-12529R2 

Predicting Firm Creation in Rural Texas: A Multi-Model Machine Learning Approach to a Complex Policy Problem 

Dear Dr. Hand:

I'm pleased to inform you that your manuscript has been deemed suitable for publication in PLOS ONE. Congratulations! Your manuscript is now with our production department. 

Kind regards, 

on behalf of

Professor Baogui Xin 

Academic Editor

PLOS ONE